# SPFormer: Enhancing Vision Transformer with Superpixel Representation

**Jieru Mei**  *meijieru@gmail.com*
*Department of Computer Science*
*Johns Hopkins University*

**Liang-Chieh Chen**  *aquariusjay@gmail.com*
*Bytedance*

**Alan Yuille**  *alan.l.yuille@gmail.com*
*Department of Computer Science*
*Johns Hopkins University*

**Cihang Xie**  *cihangxie306@gmail.com*
*Department of Computer Science and Engineering*
*University of California, Santa Cruz*

**Reviewed on OpenReview:** *https://openreview.net/forum?id=nu1SjVgSuy*

## Abstract

This work introduces SPFormer, a novel Vision Transformer architecture enhanced by superpixel representation. Addressing the limitations of traditional Vision Transformers' fixed-size, non-adaptive patch partitioning, SPFormer divides the input image into irregular, semantically coherent regions (*i.e.*, superpixels), effectively capturing intricate details. Notably, this is also applicable to intermediate features and our whole model supports end-to-end training, empirically yielding superior performance across multiple benchmarks. For example, on the challenging ImageNet benchmark, SPFormer outperforms DeiT by 1.4% at the tiny-model size and by 1.1% at the small-model size. Moreover, a standout feature of SPFormer is its inherent explainability — the superpixel structure offers a window into the model's internal processes, providing valuable insights that enhance the model's interpretability and stronger robustness against challenging scenarios like image rotations and occlusions.

## 1 Introduction

Over the past decade, the vision community has witnessed a remarkable evolution in visual recognition systems, from the resurgence of Convolutional Neural Networks (CNNs) in 2012 (Krizhevsky et al., 2012) to the cutting-edge innovation of Vision Transformers (ViTs) in 2020 (Dosovitskiy et al., 2021). This progression has instigated a significant shift in the underlying methodology for feature representation learning, transitioning from pixel-based (for CNNs) to patch-based (for ViTs).

Traditionally, pixel-based representations organize an image as a regular grid, allowing CNNs (He et al., 2016; Tan & Le, 2019; Mehta & Rastegari, 2022; Sandler et al., 2018) to extract local, detailed features through sliding window operations. Despite the inductive bias inherent in CNNs, like translation equivariance, aiding their success in effectively learning visual representations, these networks face a challenge in capturing global-range information, typically necessitating the stacking of multiple convolutional operations and/or additional operations (Li et al., 2020; Chen et al., 2018) to enlarge their receptive fields.

In contrast, ViTs (Dosovitskiy et al., 2021) regard an image as a sequence of patches. These patch-based representations, usually of a much lower resolution compared to their pixel-based counterparts, enable

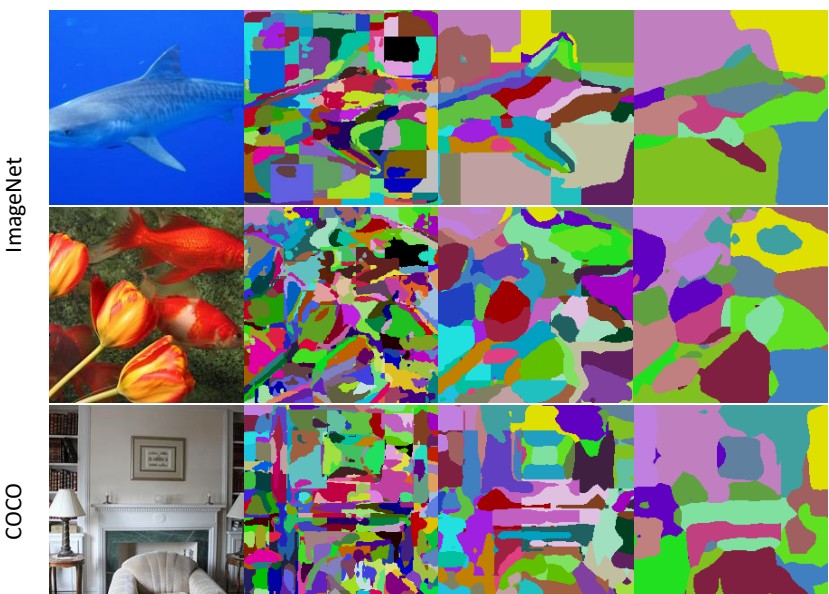

Figure 1: Visualization of learned superpixels with our SPFormer trained on ImageNet with category labels only. For each row, we show input image, visualization of 196, 49, and 16 superpixels. The learned superpixel aligns well with the object boundaries even with 16 superpixels. The last row shows results from a COCO image (not trained), demonstrating SPFormer's zero-shot ability.

global-range self-attention operations in a computationally efficient manner. While the attention mechanism successfully captures global interactions, it does so at the expense of losing local details, like object boundaries. Moreover, the low resolution of patch-based representations poses challenges to adaptation for high-resolution dense prediction tasks such as segmentation and detection, which require both local detail preservation and global context information.

This dichotomy raises an intriguing question: *can we derive benefits from both preserved local details and effective long-range relationship capture*? In response, we explore superpixel-based solutions, which have been employed extensively in computer vision prior to the deep learning era (Zhu & Yuille, 1996; Shi & Malik, 2000; Martin et al., 2001; Malik et al., 2001; Borenstein & Ullman, 2002; Tu & Zhu, 2002; Ren & Malik, 2003). These solutions provide locally coherent structures and reduce computational overhead compared to pixel-wise processing. Specifically, adaptive to the input, superpixels partition an image into irregular regions, with each region grouping pixels with similar semantics. This approach allows for a small number of superpixels, making it feasible to model global interactions through self-attention.

To this end, we introduce a novel transition of ViT architecture from patch representation to superpixel representation, through our newly developed Superpixel Cross Attention (SCA). The resulting architecture, Superpixel Transformer (SPFormer), adeptly marries local detail preservation with global-range self-attention, enabling end-to-end trainability. In comparison to standard ViT architectures, SPFormer demonstrates strong enhancements across various tasks. For instance, it achieves impressive gains on the challenging ImageNet benchmark, such as 1.4% for DeiT-T and 1.1% for DeiT-S. Notably, the superpixel representation in SPFormer aligns seamlessly with semantic boundaries, even in unseen data. Moreover, the interpretability provided by our superpixel representation facilitates understanding of the model's decision-making process and enhances the robustness against rotations and occlusions. These findings highlight the potential of superpixel-based approaches in advancing the field, and, hopefully, could inspire future research beyond traditional pixel and patch-based paradigms in visual representation learning.

## 2 Related Work

**Pixel Representation.** CNNs (LeCun et al., 1998; Krizhevsky et al., 2012; Simonyan & Zisserman, 2015; Szegedy et al., 2015; Ioffe & Szegedy, 2015; He et al., 2016; Tan & Le, 2019; Liu et al., 2022) process an

image as a grid of pixels in a sliding-window manner. CNN has been the dominant choice since the advent of AlexNet (Krizhevsky et al., 2012), benefiting from several design choices, such as translation equivariance and the hierarchical structure to extract multiscale features. However, it requires stacking of several convolution operations to capture long-range information (Simonyan & Zisserman, 2015; He et al., 2016), and it could not easily capture global-range information, as the self-attention operation (Vaswani et al., 2017).

**Patch Representation.** The self-attention mechanism (Bahdanau et al., 2015) of Transformer architectures (Vaswani et al., 2017) effectively captures long-range information. However, its computation cost is quadratic to the number of input tokens. ViTs (Dosovitskiy et al., 2021) alleviate the problem by tokenizing (or patchifying) the input image with a sequence of patches (*e.g.*, patch size $16 \times 16$). Patch representation (Dosovitskiy et al., 2021) unleashes the power of Transformer architectures (Vaswani et al., 2017) in computer vision, significantly impacting multiple visual recognition tasks (Russakovsky et al., 2015; Carion et al., 2020; Zhu et al., 2021; Wang et al., 2021a; Touvron et al., 2021a; Radford et al., 2021; Bao et al., 2022; He et al., 2022; Yu et al., 2022c; 2023). Due to the lack of built-in inductive biases as in CNNs, learning with ViTs requires special training enhancements, *e.g.*, large-scale datasets (Sun et al., 2017), better training recipes (Touvron et al., 2021a; Steiner et al., 2021), or architectural designs (Liu et al., 2021; Wang et al., 2021b). To mitigate the issue, some work exploits convolutions (LeCun et al., 1998; Sandler et al., 2018) to tokenize images, resulting in hybrid CNN-Transformer architectures (Wu et al., 2021; Yuan et al., 2021; Dai et al., 2021; Xiao et al., 2021; Mehta & Rastegari, 2022; Guo et al., 2022; Tu et al., 2022; Yang et al., 2023). Unlike those works that simply gather knowledge from existing CNNs and ViTs, we explore a different superpixel representation in ViTs.

**Superpixel Representation.** Before the deep learning era, superpixel is one of the most popular representations in computer vision (Zhu & Yuille, 1996; Shi & Malik, 2000; Martin et al., 2001; Malik et al., 2001; Borenstein & Ullman, 2002; Tu & Zhu, 2002; Ren & Malik, 2003). Ren and Malik (Ren & Malik, 2003) preprocess images with superpixels that are locally coherent, preserving the structure necessary for the following recognition tasks. It also significantly reduces the computation overhead, compared to the pixel-wise processing. The superpixel clustering methods include graph-based approaches (Shi & Malik, 2000; Felzenszwalb & Huttenlocher, 2004), mean-shift (Comaniciu & Meer, 2002; Vedaldi & Soatto, 2008), or k-means clustering (Lloyd, 1982; Achanta et al., 2012). Thanks to its effective representation, recently some works attempt to incorporate clustering methods into deep learning frameworks (Jampani et al., 2018; Yang et al., 2020; Locatello et al., 2020; Xu et al., 2022; Yu et al., 2022a; Zhang et al., 2022; Yu et al., 2022b; Ma et al., 2023; Huang et al., 2023; Zhu et al., 2023). For example, SSN (Jampani et al., 2018) integrates the differentiable SLIC (Achanta et al., 2012) to CNNs, allowing end-to-end training. Yu et al. (2022a;b) regard object queries (Carion et al., 2020; Wang et al., 2021a) as cluster centers in Transformer decoders (Vaswani et al., 2017). SViT (Huang et al., 2023) clusters the tokens to form the super tokens, where the clustering process has no gradient passed through[1]. Consequently, their network is not aware of the clustering process and could not recover from the clustering error. CoCs (Ma et al., 2023) groups pixels into clusters, while aggregating features within each cluster by regarding the image as a set of points with coordinates concatenated. In contrast, our proposed method groups pixels into superpixels, and models their global relationship via self-attention. Furthermore, during clustering, CoCs uses a Swin-style window partition (Liu et al., 2021) that introduces visual artifacts, especially around the window boundaries.

## 3 Method

We introduce SPFormer, a novel image processing framework that integrates superpixel-based feature representation with a SCA mechanism. This approach effectively addresses the limitations of traditional pixel and patch-based methods by enhancing both computational efficiency and detail preservation. We begin with a detailed discussion of our advanced superpixel representation (Sec. 3.1), followed by an explanation of the SCA mechanism in Sec. 3.2, which refines this representation by blending local details with global contextual information efficiently. Finally, we elaborate on the integration of these advancements into the SPFormer architecture in Sec. 3.3.

---

[1]From official code: `https://github.com/hhb072/STViT/blob/main/models/stvit.py#L206`

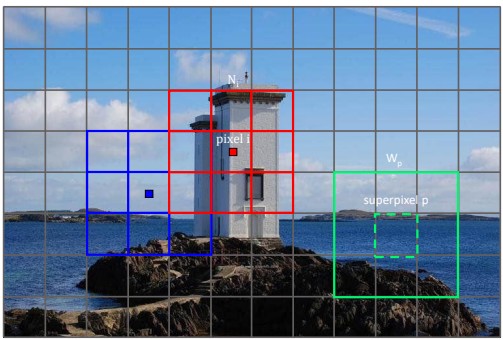

Figure 2: Illustration the relationships between a pixel $i$ and its neighboring superpixels $\mathcal{N}_i$, as well as the local window $\mathcal{W}_p$ of a superpixel $p$. In this framework, each pixel $i$ is associated with adjacent superpixels within a neighborhood radius of $r$. For the purposes of this study, we set $r = 3$. Conversely, each superpixel $p$ incorporates a set of pixels that fall within its local window $\mathcal{W}_p$. These pixels are selected based on their proximity and linkage to the superpixel $p$, as determined by their membership in $\mathcal{N}_i$.

### 3.1 Superpixel Representation: Bridging Pixel and Patch Approaches

In the evolving landscape of feature representation, the transition from pixel to patch-based methods in ViTs has opened new avenues for image processing. However, each method has inherent limitations, inspiring our exploration of a more adaptive and efficient representation: superpixels.

**Pixel Representation.** Traditional pixel representation conceptualizes an image **I** as a grid of high-resolution pixels, expressed as $\mathbf{I} \in \mathbb{R}^{c \times h \times w}$. Predominantly employed in CNN-based methods, this approach suffers from restricted contextual integration due to the limited receptive fields of the convolution operations. Although self-attention mechanisms could theoretically mitigate this limitation, their implementation is computationally prohibitive at this scale due to the quadratic dependencies on the number of pixels.

**Patch Representation.** Contrastingly, ViTs adopt a patch-based representation with reduced resolution, denoted by $\mathbf{P} \in \mathbb{R}^{c \times p_h \times p_w}$. This simplification reduces computational complexity and facilitates the application of self-attention mechanisms—albeit at a loss of finer detail and contextual richness due to the coarse granularity.

**Superpixel Representation.** Bridging the benefits of both pixel and patch-based approaches, our model employs a superpixel representation, encompassing features $\mathbf{S} \in \mathbb{R}^{c \times s_h \times s_w}$ and an association matrix $\mathbf{A} \in \mathbb{R}^{n \times h \times w}$. The employment of superpixels significantly reduces the number of feature components; for instance, in this study, we utilize a stride of $s = h/s_h = w/s_w = 4$, effectively leading to a reduction by a factor of 16 relative to the original pixel count. This association matrix establishes a competitive relationship among pixels and their corresponding superpixels, an aspect that will be elaborated in ensuing detailed discussions on the structured components of our approach.

1. *Neighboring Superpixels* ($\mathcal{N}_i$): Each pixel $i$ is connected to its neighboring superpixels, comprising the nearest superpixel and those within its Moore Neighborhood, as shown in Fig. 2. Specifically, the neighborhood radius is $r$, resulting in a total of $n = r^2$ adjacent superpixels. In the context of this study, we set $r$ to 3, thereby involving 9 superpixels around each pixel. This configuration not only facilitates localized information sharing but also enhances the granularity of the feature representation. Moreover, these superpixels engage in a competitive interaction to establish dominance over the association with the pixel, which dynamically affects the feature integration process.

2. *Superpixel's Local Window* ($\mathcal{W}_p$): For each superpixel $p$, a local window $\mathcal{W}_p$ is defined, which encompasses pixels that are potentially owned by the superpixel, based on the $\mathcal{N}_i$ associations. This window configuration is critical for implementing the superpixel cross attention mechanism via a sliding window technique, as detailed in Sec. 3.2. It ensures that each superpixel contextually interacts with its relevant local pixel environment, enabling precise and adaptive refinement of attention dynamics.

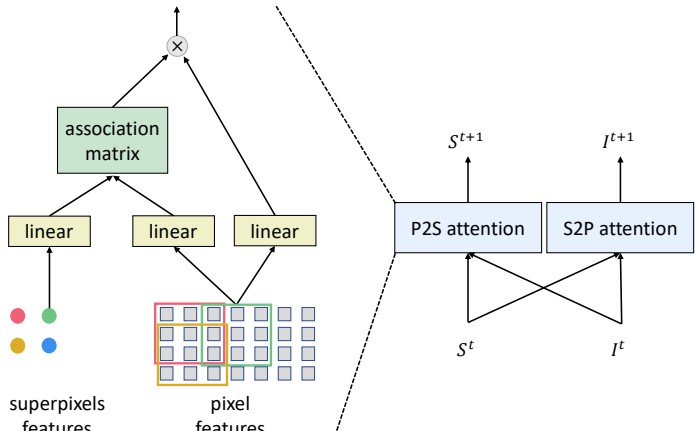

Figure 3: Illustration of our SCA module, demonstrating iterative refinement of superpixel and pixel features using a sliding window-based cross-attention mechanism. Each superpixel cross-attends to a localized region of pixels, as highlighted in the colored rectangle. On the left, we detail the P2S cross-attention process, while the S2P cross-attention is depicted similarly, albeit with reversed roles for superpixel and pixel.

The superpixel representation can be effectively converted into a pixel representation using the equation:

$$\mathbf{I}_i = \sum_{p \in \mathcal{N}_i} \mathbf{A}_{ip} \cdot \mathbf{S}_p \tag{1}$$

In Eq. 1, $\mathbf{I}_i$ represents the feature vector corresponding to the pixel $i$, while $\mathbf{S}_p$ represents the feature vector of the superpixel $p$. The term $\mathbf{A}_{ip}$ denotes the association weight between the pixel $i$ and the superpixel $p$. This transformation method not only preserves boundary information with high fidelity but also offers a finer granularity than direct patch upsampling. It facilitates a seamless transition between superpixel and pixel representations, enabling the use of a more manageable superpixel grid while ensuring meticulous retention of detailed pixel-level information.

Furthermore, our superpixel approach effectively retains crucial boundary information and demonstrates robustness against common image distortions such as rotation and occlusion, benefiting from the inherent adaptability of superpixels. The reduced computational demands, along with semantic pixel clustering for improved explainability, and resilience against complex transformations, position this method as a superior alternative to traditional approaches.

## 3.2 Superpixel Cross Attention

The SCA module utilizes a dual attention mechanism: Pixel-to-Superpixel (P2S) and Superpixel-to-Pixel (S2P). This design aims at localized enhancement of feature representations, critical for applications demanding high-resolution detail and precise contextual understanding (Fig. 3).

### 3.2.1 Mechanisms of SCA

Both P2S and S2P mechanisms are engineered to optimize the interplay between pixel and superpixel features:

- **P2S Cross-Attention**: This process enhances the representation of superpixels by integrating contextual details from pixels within specific local windows.

- **S2P Cross-Attention**: Conversely, this attention refines pixel features by drawing from the contextual data of adjacent superpixels, thereby enriching the detail and granularity of pixel features.

These bidirectional processes effectively bridge superpixel efficiency and the detailed granularity of pixel representations, offering a versatile and refined feature set.

### 3.2.2 Iterative Feature Refinement

Starting with initial pixel $\mathbf{I}^0 \in \mathbb{R}^{c \times h \times w}$ and superpixel features $\mathbf{S}^0 \in \mathbb{R}^{s_c \times s_h \times s_w}$, each iteration $t$ progressively refines these features. Updates are applied to both superpixel features $\mathbf{S}^t$ and association matrix $\mathbf{A}^t$ to ensure continual improvement of the superpixel representation.

In the S2P cross-attention phase, starting with pixel features $\mathbf{I}^{t-1}$ and superpixel features $\mathbf{S}^{t-1}$, we first recreate the association $\mathbf{A}_{ip}^t$ between pixel $i$ and superpixel $p$, as demonstrated:

$$\mathbf{A}_{ip}^t = \text{softmax}_{p \in \mathcal{N}_i} \left( \mathbf{q}_{\mathbf{I}_i^{t-1}} \cdot \mathbf{k}_{\mathbf{S}_p^{t-1}} \right) \tag{2}$$

where $\mathcal{N}_i$ comprises the neighboring superpixels of pixel $i$, as defined in Sec. 3.1, and $\mathbf{q}$, $\mathbf{k}$, and $\mathbf{v}$ symbolize the query, key, and value vectors derived from linear transformations of the previous iteration's features. Building upon Eq. (1), pixel updates are generated by mapping superpixel features $\mathbf{v}_{\mathbf{S}_p^{t-1}}$ back to pixels using the freshly computed associations $\mathbf{A}_{ip}^t$ in a residual manner:

$$\mathbf{I}_i^t = \mathbf{I}_i^{t-1} + \sum_{p \in \mathcal{N}_i} \mathbf{A}_{ip}^t \cdot \mathbf{v}_{\mathbf{S}_p^{t-1}} \tag{3}$$

Concurrently, superpixel features are updated through P2S cross-attention, as formulated below:

$$\mathbf{S}_p^t = \mathbf{S}_p^{t-1} + \sum_{i \in \mathcal{W}_p} \text{softmax}_i \left( \mathbf{q}_{\mathbf{S}_p^{t-1}} \cdot \mathbf{k}_{\mathbf{I}_i^{t-1}} \right) \mathbf{v}_{\mathbf{I}_i^{t-1}} \tag{4}$$

This equation iterates over pixels i located within the local window $\mathcal{W}_p$, defined in Sec. 3.1.

These differentiable update equations are essential for refining both pixel and superpixel feature representations, enabling the gradients from either domain to be leveraged for end-to-end training.

### 3.2.3 Positional Embedding and Multi-head Attention

To enhance the spatial fidelity of superpixels, positional information is integrated into the SCA module using Convolution Position Embedding (CPE) (Huang et al., 2023), which captures spatial relationships within the image. Both superpixel and pixel features are augmented with CPE implemented as a $3 \times 3$ depthwise convolution supplemented with a skip connection, applied prior to the execution of P2S and S2P cross-attention mechanisms. This augmentation further encourages the locality of the superpixels.

Given the propensity for over-segmentation in superpixel methods, multiple segmentations can emerge; thus, relying exclusively on a single superpixel segmentation may not suffice for an exhaustive image description. Contrary to prior methodologies such as those proposed by Jampani et al. (2018) and Zhu et al. (2023), which generate a singular type of superpixel segmentation at a time, our approach within the SCA module incorporates multi-head attention. This technique facilitates intricate interactions between superpixels and pixels by leveraging global contextual information. As such, it produces a spectrum of superpixel representations, depicted in Fig. 5. These varied representations address distinct levels of granularity and ambiguity associated with superpixel over-segmentation, thus enhancing the method's utility and adaptability.

### 3.3 SPFormer Architecture

Our architecture capitalizes on the sophisticated capabilities of our superpixel representation while adhering closely to the foundational structure of the standard ViT as delineated by Dosovitskiy et al. (2021), thus facilitating a direct and fair comparison.

Unlike the standard ViT that regards the images as $16 \times 16$ patches, we utilize a more granular approach for extracting initial pixel features $\mathbf{I}^0 \in \mathbb{R}^{c \times h \times w}$, opting for a lightweight stem with a stride of 4. This modification includes a conventional stride 4 non-overlapping patchify layer. Additionally, in variants of our

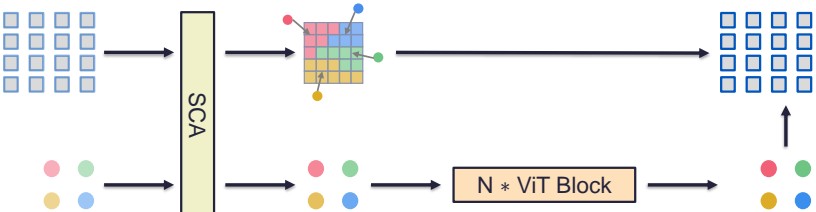

Figure 4: Illustration of a single stage of the SPFormer architecture. It starts with initial superpixel features and pixel features as inputs. The SCA module iteratively refines superpixel features, enhancing their semantic richness. These features are then processed by the MHSA for global contextual understanding. The stage concludes by updating the pixel features based on the enriched superpixel information. These updated pixel and superpixel features are either fed into the subsequent stage, which starts with another SCA module, or utilized for making final predictions.

model, denoted by $^\dagger$, the patchify layer is replaced by two convolutional layers each with a kernel size of 3 and a stride of 2. This convolutional stem provides a more effective initialization of pixel features, aiding in better feature representation from the onset.

For the extraction of superpixel features, we implement a stride $s = 4$, producing superpixel features $\mathbf{S}^0 \in \mathbb{R}^{s_c \times s_h \times s_w}$ directly from the pixel features $\mathbf{I}^0$. This process involves a $1 \times 1$ convolution followed by a $4 \times 4$ average pooling operation. With this configuration, the resultant number of superpixels is equivalent to the number of patches in the standard ViT. Such equivalence crucially ensures that the computational costs associated with the remainder of our network align closely with those of the classical ViT framework, apart from the minimal, yet strategically important, overhead introduced by the integrated Superpixel Cross Attention (SCA) layers.

As shown in Fig. 4, $\mathbf{S}^0$ and $\mathbf{I}^0$ undergo continuous refinement across multiple iterations $t = 2$ using the SCA module. This module enhances the semantic alignment of the superpixel representation by exploiting the local spatial context of each superpixel, as detailed in Sec. 3.2. Subsequently, the enriched superpixel features $\mathbf{S}^t$ are processed by multi-head self-attention to extract long-range dependencies and contextual nuances within the image.

We observed that even after multiple iterations (e.g., $t > 2$), the superpixel representations generated by a single SCA module may not fully align with the overarching context. This misalignment primarily stems from insufficient semantic information derived from the output of the lightweight stem. To tackle this challenge, we implement a gradual refinement strategy for enriching superpixel representations by interleaving multiple SCA modules with the ViT Blocks. We segment the original ViT architecture into multiple stages, each starting with a SCA module with $t = 2$ iterations). This design allows each SCA module to build on the semantically enriched superpixel features from the previous stage, progressively enhancing the semantic depth.

Specifically, at the end of the stage, superpixel features enriched with global attention are projected back into the pixel domain, as outlined in Eq. (1), and then added with the existing pixel features $\mathbf{I}^t$. This integration process ensures that the pixel features are augmented with comprehensive global contextual information. In the subsequent stage, these contextually enriched pixel and superpixel features are utilized as inputs, avoiding the need to reinitialize features from scratch. This continuous, integrative approach not only aligns the superpixels more closely with low-level boundaries but also enhances consistency with semantic contours, thereby improving the overall coherence and relevance of the image representation.

Conceptually, our network architecture is designed as a dual-branch structure. One branch preserves a dense pixel representation with high resolution, while the other branch focuses on our compact superpixel representation. Minimal direct operations are applied to the dense pixel representation, enabling us to allocate most computational resources to the more efficient superpixel representation. This dual-branch approach fosters computational efficiency without sacrificing the detailed preservation of local image features.

By integrating superpixel representation with SCA, SPFormer offers an advanced solution for boosting both the semantic interpretability and computational efficiency of image processing. This combination enhances detail retention and positions SPFormer as a strong tool in diverse image processing applications.

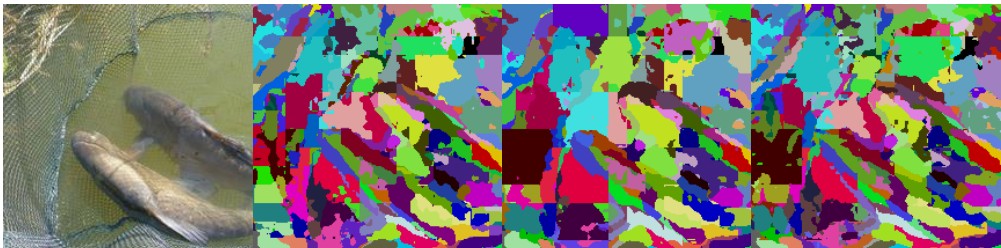

Figure 5: The multi-head SCA design generates multiple superpixel representations, each capturing different semantic relationships and addressing the ambiguity in superpixel over-segmentation.

## 4 Experiments

### 4.1 Implementation Details

Different number of heads of the attentin are used to enable multiple superpixel segmentation in a SCA module. For our architecture variants, we allocate two attention heads for the smaller models (SPFormer-T and SPFormer-S) and three heads for the base model (SPFormer-B).

As outlined in Sec. 3.3, SCA blocks are interleaved into the standard ViT architecture, specifically before the first and third self-attention blocks. We utilize the LayerScale technique, as specified in Touvron et al. (2021b), to modulate gradient flow and enhance training stability and effectiveness. With the inclusion of residual connections as detailed in Eq. (4) and Eq. (3), our approach starts by mimicking standard patches but evolves to exploit superpixels more effectively as training advances.

Adhering to the protocols established in DeiT (Touvron et al., 2021a), we implement robust data augmentations, use the AdamW optimizer, and follow a cosine decay learning rate schedule. All models train on the ImageNet dataset (Russakovsky et al., 2015) for 300 epochs. During SPFormer-B/16 training, significant overfitting challenges arose. Increasing the Stochastic Depth (Huang et al., 2016) rate from 0.1 to 0.6 effectively addressed these issues, highlighting a potential need for advanced regularization strategies tailored to superpixel representation for future studies.

### 4.2 Efficiency in Image Classification and Segmentation

#### 4.2.1 Main Results on ImageNet

Our evaluation of SPFormer on the ImageNet dataset demonstrates its superior efficiency and performance over the DeiT baseline under varying configurations as shown in Tab. 1. Specifically, SPFormer-S, which employs the standard ViT configuration with 196 tokens, exceeds the performance of DeiT-S by 1.1%, achieving a top-1 accuracy of 81.0% compared to 79.9% for DeiT-S. Furthermore, SPFormer-T outperforms DeiT-T by 1.4%, recording 73.6% versus 72.2%.

The superior detail preservation afforded by superpixel representation enables SPFormer to exhibit scaling behaviors distinct from those of ViT, which utilizes patch representation. This advantage becomes increasingly evident with larger patch sizes, such as 32. While DeiT-S/32 suffers from performance degradation due to its coarser granularity, SPFormer-S/32 demonstrates stable performance, achieving 76.4%, which is a notable 4.2% improvement over DeiT-T and requires fewer FLOPs.

The token reduction capability of the superpixel representation permits a strategic focus on increasing image resolution to capture finer details effectively. By increasing the stride $s$ from 4 to 8 and adopting a higher resolution of 448 which produces the same number of of superpixels, SPFormer achieves a 0.3% enhancement in performance relative to its standard SPFormer-S configuration, without incurring extra computational costs. Conversely, similar adjustments in DeiT-S only yield a negligible improvement of 0.1%, hindered by the granularity of its patch-based representation.

Employing a convolutional stem provides an enhanced initialization for pixel features, which subsequently leads to improved representations of superpixels in the initial stage. This modification has been shown to

Table 1: Comparative analysis of SPFormer's performance on ImageNet classification against DeiT baselines. Variants augmented with two convolution layers of kernel size 3 with stride 2 are denoted by $^\dagger$.

| Model | #Params | #FLOPs | Top-1 |
|---|---|---|---|
| SPFormer-S/56 | 22M | 0.5G | 72.3 |
| DeiT-T (Touvron et al., 2021a) | 5M | 1.3G | 72.2 |
| SPFormer-T | 5M | 1.3G | 73.6 |
| SPFormer-T$^\dagger$ | 5M | 1.3G | 75.0 |
| DeiT-S/32 (Touvron et al., 2021a) | 22M | 1.1G | 73.3 |
| SPFormer-S/32 | 22M | 1.2G | 76.4 |
| SPFormer-S/32$^\dagger$ | 22M | 1.3G | 77.9 |
| DeiT-S (Touvron et al., 2021a) | 22M | 4.6G | 79.9 |
| SPFormer-S | 22M | 5.2G | 81.0 |
| SPFormer-S$^\dagger$ | 22M | 5.3G | 81.7 |
| DeiT-B (Touvron et al., 2021a) | 87M | 17.5G | 81.8 |
| SPFormer-B | 87M | 19.2G | 82.4 |
| SPFormer-B$^\dagger$ | 87M | 19.2G | 82.7 |

Table 2: Ablation study on the design choices in SPFormer.

| Model | #Params | #FLOPs | Top-1 |
|---|---|---|---|
| SPFormer-S/32 | 22M | 1.2G | 76.4 |
| Single Iteration in SCA | 22M | 1.2G | 75.4 |
| SCA at Initial Layer Only | 22M | 1.2G | 74.8 |
| Single-Head SCA | 22M | 1.2G | 75.6 |
| Learnable Position Embedding | 22M | 1.2G | 76.1 |

consistently enhance performance, as exemplified by SPFormer-S/32$^\dagger$. This particular configuration achieved a notable increase of 1.5% in ImageNet classification accuracy, attaining a total of 77.9%. These findings further underscore the significance of precise superpixel assignment and demonstrate its advantages over the traditional vanilla patch representation. Such enhancements validate the efficacy of our modified approach in handling complex image processing tasks more effectively.

### 4.2.2 Ablation Study: Design Choices in SPFormer

We evaluates key design elements of SPFormer-S/32 on the ImageNet validation set. We investigate the impacts of iteration count in the SCA module, the placement of SCA within the architecture, the use of multi-head attention, and the choice of position embeddings.

The findings highlight the importance of multiple iterations in SCA for performance enhancement, with a single iteration leading to a 1.0% drop in accuracy. The strategic placement of SCA across different layers is crucial, as restricting it to the initial layer causes a 1.6% accuracy reduction, indicating that higher-level features play a vital role in augmenting semantic depth and in rectifying early-stage superpixel inaccuracies. Furthermore, employing multi-head attention in SCA is significant for capturing diverse superpixel relationships, with its absence leading to a 0.8% decrease in accuracy. Lastly, using learnable position embeddings over CPE results in a slight drop in performance.

This ablation study validates the effectiveness of the considered design choices in SPFormer, affirming their contributions to the overall performance of the model.

### 4.2.3 Semantic Segmentation: Utilizing SPFormer's High-Resolution Feature Preservation

SPFormer's superpixel representation intrinsically maintains higher resolution features, making it particularly suitable for semantic segmentation tasks. This characteristic allows for detailed and context-rich segmentation outputs, a key advantage over traditional patch-based methods.

Table 3: Semantic segmentation on ADE20K val split.

| Method | #Params | #FLOPs | Pretrained | mIoU |
|---|---|---|---|---|
| DeiT-S | 22M | 32G | ✗ | 20.1 |
| SPFormer-S | 23M | 35G | ✗ | 23.1 |
| DeiT-S | 22M | 32G | ✓ | 42.3 |
| SPFormer-S | 23M | 35G | ✓ | 46.5 |

Table 4: Semantic segmentation on Pascal Conext val split.

| Method | #Params | #FLOPs | Pretrained | mIoU |
|---|---|---|---|---|
| DeiT-S | 22M | 27G | ✗ | 18.0 |
| SPFormer-S | 23M | 30G | ✗ | 21.1 |
| DeiT-S | 22M | 27G | ✓ | 48.3 |
| SPFormer-S | 23M | 30G | ✓ | 51.2 |

Incorporating SPFormer into the SETR (Zheng et al., 2021) framework, we enhance segmentation performance by directly classifying individual superpixels. This direct approach leverages SPFormer's high-resolution feature preservation, allowing for more nuanced segmentation. The final segmentation maps are generated by upscaling the superpixel-based logits using Eq. (1).

We evaluate SPFormer on the ADE20K (Zhou et al., 2017) and Pascal Context (Mottaghi et al., 2014) datasets. Utilizing ImageNet-pretrained models, SPFormer demonstrates significant improvements in mIoU, highlighting its effectiveness in detailed segmentation tasks.

As shown in Tab. 3 and Tab. 4, the performance gains in mIoU are noteworthy when using ImageNet-pretrained models: 4.2% improvement on ADE20K and 2.8% on Pascal Context. These results not only highlight the detailed nature of SPFormer's superpixel representation but also its adaptability to diverse and complex datasets. To further validate the intrinsic segmentation capabilities of SPFormer, we conduct additional training from scratch. This approach reiterates the model's strength in maintaining high-resolution features independently of pretraining influences, leading to mIoU improvements of 3.0% on ADE20K and 3.1% on Pascal Context compared to baseline methods.

### 4.3 Unveiling SPFormer's Explainability

Integrating superpixel representation into the Vision Transformer architecture adds a significant layer of explainability compared to conventional fixed patch partition methods. This section first discusses the inherent explainability of our superpixel representation, followed by an evaluation of its semantic alignment and generalizability to unseen data.

#### 4.3.1 Superpixel Representation as an Explainability Tool

Our method's superpixel representation can be visualized through the association matrix $\mathbf{A}$, providing insights into the model's internal processing. In Fig. 1, we visualize the learned soft associations by selecting the argmax over the superpixels:

$$\hat{\mathbf{A}} = \mathrm{argmax}(\mathbf{A}) \tag{5}$$

These visualizations reveal that, even with a soft association, the superpixels generally align with image boundaries. This alignment is noteworthy as it emerges even though the network is only trained on image category labels. Thus, the model segments images into irregular, semantics-aware regions while reducing the number of tokens needed for representation.

Furthermore, we assess the generalizability of our superpixel representation using the COCO dataset (Lin et al., 2014), which consists of high-resolution images with complex scenes. For this evaluation, we resize and center-crop COCO images to align with the ImageNet evaluation pipeline. Fig. 6 showcases the

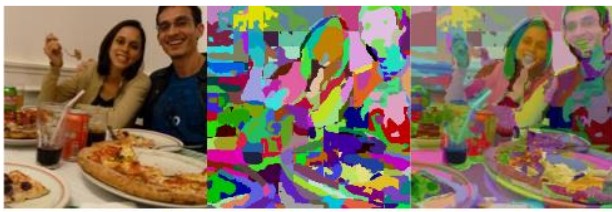 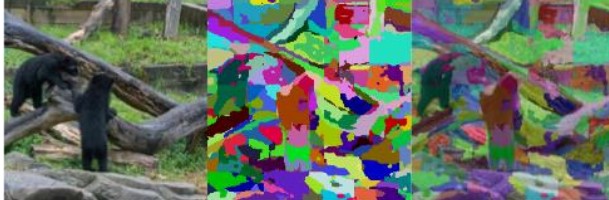

Figure 6: Zero-shot transferability on the COCO dataset. Trained solely on ImageNet, SPFormer demonstrates effective segmentation of unseen COCO images into detailed superpixels. 196 superpixels are used in this visualization.

Table 5: Evaluation of superpixel quality in a zero-shot setting on Pascal VOC 2012 and Pascal-Parts-58 datasets, using 196 patches/superpixels. Our SPFormer variants demonstrate notable improvements over traditional patch representations and are competitive with the SLIC method.

| Method | Pascal Voc2012 | | Pascal-Parts-58 | |
|---|---|---|---|---|
| | mIoU | mAcc | mIoU | mAcc |
| Patch | 87.8 | 92.8 | 68.7 | 78.2 |
| SPFormer-T$^\dagger$ | 91.5 | 95.7 | 71.5 | 79.9 |
| SPFormer-S$^\dagger$ | 92.0 | 96.6 | 73.3 | 82.4 |
| SPFormer-B$^\dagger$ | 91.2 | 96.3 | 72.5 | 81.4 |
| SLIC (Achanta et al., 2012) | 92.5 | 95.4 | 74.0 | 81.7 |

visual representation of superpixels on these images. Note that the superpixels generated by SPFormer, trained exclusively on ImageNet, adapt well to this unseen data, capturing intricate structures such as thin objects. This adaptability highlights the model's capability to preserve detail and generalize its superpixel representation to new contexts.

### 4.3.2 Semantic Alignment of Superpixels

Our evaluation of superpixel representation focuses on its ability to align with ground truth boundaries in images, despite the model not being trained on the datasets used for this assessment. This test involved a quantitative analysis on both object and part levels using the Pascal VOC 2012 dataset (Everingham et al., 2015) and Pascal-Part-58 (Zhao et al., 2019). Note that these assessments were performed without any training on these specific datasets, underscoring the model's generalization capabilities.

In our approach, each superpixel or patch's prediction is derived by aggregating the ground truth labels of the pixels it encompasses. We assign the most frequently occurring label within a superpixel as its prediction, assuming optimal classification. This method leverages the soft associations produced by our SCA module, where predictions are formed by combining pixel labels with their corresponding weights and upscaled as per Eq. (1).

Diverging from the single-superpixel outcome of traditional patch representation, our model employs a multi-head design in the SCA module. This allows for the creation of multiple, distinct superpixels for each head, enhancing the richness and diversity of the extracted features (see Fig. 5). For our evaluations, we computed an average of predictions across all heads. It is noteworthy that effective feature extraction in our model is deemed successful if even a single head accurately identifies a superpixel.

The proficiency of our superpixel approach is demonstrated in its performance compared to vanilla ViTs that utilize patch representations with a stride of 16. ViTs often suffer from a granularity trade-off, losing finer details in favor of broader patch representations. In contrast, the superpixels from our SCA module, as shown in Tab. 5, manifest substantial improvements — achieving a 4.2% increase in object-level and 4.6% in part-level mIoU with SPFormer-S$^\dagger$. Furthermore, these superpixels display a quality comparable to those from traditional superpixel methods like SLIC (Achanta et al., 2012), highlighting our method's effectiveness in capturing detailed semantic information without direct training on the evaluation datasets.

Table 6: Quantitative evaluation of SPFormer's robustness to rotation, comparing performance at different angles. Variants augmented with two convolution layers of kernel size 3 with stride 2 are denoted by [†].

| Model | Clean | 15 | 30 | 45 |
|---|---|---|---|---|
| DeiT-S/32 (Touvron et al., 2021a) | 73.3 | 71.1 | 67.7 | 59.5 |
| SPFormer-S/32[†] | 77.9 | 75.2 | 73.4 | 66.9 |

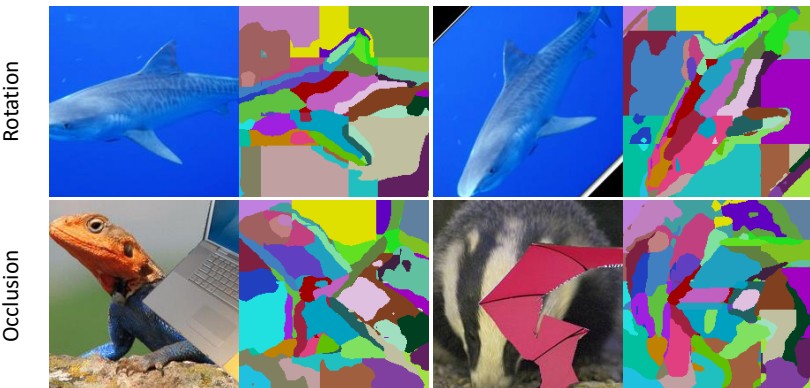

Figure 7: Visualization of SPFormer's superpixel representation under rotation and occlusion, highlighting the model's adaptability and robustness.

### 4.3.3 Explainability-Driven Robustness

The robustness of SPFormer is deeply intertwined with its explainability, particularly through the superpixel representation. This section explores how the model's transparent and interpretable features contribute to its resilience against image modifications like rotation and occlusion.

**Robustness to Rotation.** Our model's capacity to generate coherent superpixels even under rotational transformations showcases the robustness afforded by its explainable structure. For example, Fig. 7 illustrates the model's performance under rotation, and Tab. 6 quantifies this robustness, under patch size 32. By visualizing how superpixels adapt to rotated images, we gain insights into the model's stability in varied orientations. While SPFormer demonstrates a heightened robustness to rotation, it still exhibits some limitations, likely due to the learnable absolute position embeddings not being inherently rotation-invariant. These observations suggest potential avenues for enhancing rotational robustness, possibly through integrating rotation-invariant mechanisms within the superpixel representation or the network architecture.

**Robustness to Occlusion.** The occlusion robustness of SPFormer is another facet where explainability plays a key role. By examining superpixel behavior in occluded images, we observe the model's ability to distinguish between occluders and the object of interest. Unlike traditional patch-based representations, which tend to blend occluders with the object, our superpixel representation more effectively isolates and identifies obscured parts of the image. This nuanced differentiation is a direct result of the model's explainable superpixel structure, which provides a more detailed and context-aware interpretation of the image content, as demonstrated in Fig. 7.

## 5 Conclusion

In this work, we introduced SPFormer, a novel approach for feature representation in Vision Transformers, emphasizing superpixel representation. This method showcases a promising shift from traditional pixel and patch-based approaches, offering two distinct advantages: efficiency due to a reduced number of superpixels facilitating global self-attention and explainability through semantic grouping of pixels. The empirical results demonstrate the potential of SPFormer in diverse computer vision tasks, under both the standard benchmarks and challenging recognition scenarios. We hope our findings can pave the way for future exploration in this direction, encouraging further research into superpixel-based visual representation learning.

## Acknowledgment

This work was supported by ONR N00014-21-1-2690. We thank the Center for AI Safety for supporting our computing needs.

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
