# OpenReview forum: "SPFormer: Enhancing Vision Transformer with Superpixel Representation"
_TMLR — Accepted by TMLR_

### Review · Reviewer_MvDf · 2024-10-08

**Summary Of Contributions:**

This paper proposes incorporating the superpixel representation into the vision transformer (ViT), thereby improving the patch-wise representation used by the vanilla ViT.

Implementation-wise, this work introduces the Superpixel Cross Attention (SCA) module that is prepended ahead of the traditional attention layers, which performs iterative refinements in obtaining the embeddings of the superpixels. Though, interestingly, as the SCA modules output two sets of embeddings (superpixel, and pixel embeddings), the architecture is effectively a dual-branch design.

**Audience:**

Yes

**Broader Impact Concerns:**

No broader impacts were identified.

**Claims And Evidence:**

No

**Requested Changes:**

*This section largely is a brief summary of the weaknesses part above.*

- The comparison of related work (both more state-of-the-art vision transformers such as SwinTransformer and more domain-specific works that are heavily related to the idea of superpixels) is a key missing point from my point of view. Perhaps the author could consider adding such comparisons to demonstrate that the proposed method is indeed superior.

- Likewise, the ablation studies that highlight the efficiency of the superpixels would be a strong argument and a nice thing to have -- this is especially true since the authors repeatedly state that the computational cost is a benefit of the proposed model (Section 3).

Based on the above two points, I am tentatively rating the **Claims And Evidence** as **No**.

---

- The modifications on the notations would be a relatively small improvement. Though, it will help readers understand the method better.

**Strengths And Weaknesses:**

Strengths

* The idea of leveraging superpixel seems intriguing, albeit some previous works (Jampani et al. (2018) and Zhu et al. (2023)) have studied it.
* The motivation is clearly conveyed.
* The accompanying visualizations of seemingly semantically clustered patches nicely demonstrate the strengths of the proposed superpixel patches.

Weaknesses

**The utility of superpixels, in comparison to the vanilla patch-wise representations of ViT**

My main concern is stated above. Specifically:

- If the main benefit of the superpixel embeddings is on the performance of the vision backbone, then comparing it with DeiT (2021) alone may be insufficient as of 2024, as there have been some more advanced architectures available, for instance, SwinTransformer and its variants.

- Additionally, given that the concept of superpixel has been studied before (in fact, the author has mentioned some of them in the text, e.g., Jampani et al. (2018)), it poses the question as to why the author has not compared the performance of the proposed method with such work.

- Subsequently, I would imagine another potential benefit of such arrangements is the drastic reduction in the number of tokens needed -- however, this would depend on the granularity of the embeddings. Yet, the author has not provided ablation studies on this aspect. It would be nice to do so (specifically, superpixel granularity *vs.* performance *vs.* inference cost).


**Some notations are slightly confusing**

In Section 3.1, the author introduces the superpixel representations, however:

- what will be the dimensions of $s_h$ and $s_w$ for $\mathbf{S}$? Because given the visualizations, the superpixel patches in an image will no longer be of fixed quantity (as in the vanilla ViT patches where it is simply $N\times N$). Also, is $\mathbf{A}_{i\times p} \cdot \mathbf{S}_p$ functioning as a weighted sum here -- given that in Equation 1 the pixel representation for pixel $i$ is obtained as such. Perhaps an example of the matrices would be nice (or a visualization of such transformations between pixel and superpixel representations).

---

> ### Author Response · Authors · 2024-11-11
> **Response to Reviewer MvDf**
>
> **1. what's the main benefit of superpixel representation**
>
> The primary benefit of the superpixel representation is it's efficiency, Explainability and robustness, instead of solely on the performance of the vision backbone.
>
> We chose to maintain the core architectural framework of the ViT to isolate the effects of the representation changes. By minimizing deviations from the standard ViT architecture, we ensure that any observed improvements can be attributed clearly and directly to the representation changes, facilitating a straightforward comparison.
>
> **2. Concerning the potential benefits of reduced token numbers via superpixel use depending on their granularity:**
>
> We appreciate your suggestion regarding the need for ablation studies focusing on the granularity of superpixel embeddings against performance and inference costs. As highlighted in our study, our experiments initially matched the number of superpixels to the number of patches in a standard ViT setup to precisely attribute performance improvements to the superpixel representation's intrinsic qualities, not merely to a reduction in the number of tokens.
>
> However, we acknowledge the importance of exploring how varying levels of granularity affect performance. Originally in Section 4.2.1, where we introduce configurations such as SPFormer-S/32 and SPFormer-S/32 conv stem, demonstrating respective improvements of 3.1% and 4.6% over traditional patch-based ViT models. The SPFormer-S/32 also showcases robust performance at 76.4%, which notably exceeds the performance of DeiT-T by 4.2%, while **requiring fewer computational resources**.
>
> To improve clarity and detail, we have revised the manuscript to further emphasize the scalability effects of superpixel granularity.
>
> **3. What are the dimensions of $s_h$ and $s_w$ for S? Given the visualizations, will the superpixel patches in an image maintain a fixed quantity?**
>
> We apologize for any previous confusion surrounding this issue. The number of superpixels within our framework is indeed fixed. For an image with dimensions $h \times w$, we use a constant stride $s=4$ to form a superpixel grid with dimensions $s_h \times s_w$. This setup ensures that the total number of superpixels remains consistent across different images. To aid understanding, we have added a new Figure 2 in the revised manuscript.
>
> **4. Visualization of the Association**
>
> Due to the high dimensionality of the attention matrix, $A_{ip}$, visualizing this component directly poses a significant challenge. To address this, we currently utilize Equation (5) for visualization, interpreting the matrix through its physical implications, as inspired by the work of Jampani et al. [1].
>
> [1] Varun Jampani, Deqing Sun, Ming-Yu Liu, Ming-Hsuan Yang, and Jan Kautz, "Superpixel Sampling Networks." In Vittorio Ferrari, Martial Hebert, Cristian Sminchisescu, and Yair Weiss (eds.), Proceedings of the European Conference on Computer Vision (ECCV), 2018.

---

> > ### Author Response · Authors · 2024-11-11
> > **Response to Reviewer MvDf Continued**
> >
> > **5. Comparing Superpixel Learning: This Paper vs. Previous Studies**
> >
> > Key Differences:
> >
> > 1. **Gradient Flow:**
> >    The absence of gradient flow in [2] means their network remains unaware of the clustering process, inhibiting its ability to correct clustering errors, which is crucial for adaptive learning.
> >
> > 2. **Superpixel Segmentation Ambiguity:**
> >    Given the inherent over-segmentation nature of superpixels, our method utilizes a multi-head cross-attention mechanism to produce a spectrum of possible superpixel segmentations. This approach is detailed in Section 3.2.3 of our paper and contrasts sharply with [1][2], which relies on a singular segmentation approach. Such a singular method may not capture the diverse contextual information present in complex images.
> >
> > 3. **Residual Update of Superpixel Representation:**
> >    Unlike traditional methods and [1][2], our technique employs residual updates to the superpixel features, enhancing the training stability. This approach ensures gradual updates that build intelligibly upon prior iterations, facilitating better convergence and addressing common challenges like exploding or vanishing gradients.
> >
> > 4. **Heavy Backbone and Dense Annotation Requirements:**
> >    [2] requires a full convolutional neural network (CNN) as input and dense annotations for learning superpixels, suggesting a dependency on substantial pre-existing data and substantial computational resources. In contrast, our approach with interleaved SCA layers enables the generation of superpixels from progressively refined context features augmented by ViT blocks without the need for dense annotations.
> >
> > **Performance Comparison:**
> >
> > Since the study presented in [1] is primarily focused on learning superpixel representations, which are not directly applicable to classification tasks, our experimental comparisons have been mainly conducted using the methodologies and configurations outlined in [2].
> >
> > To empirically validate the theoretical differences highlighted, we incorporated [1]’s superpixel representation into our SPFormer-S/32 model for a direct performance comparison using the official released code. The substitution of our superpixel representation with that from [1] led to a notable decrease in model accuracy, as detailed in the table below. This underperformance is likely attributable to the disabled gradient flow in [1]'s method, which prevents the network from effectively correcting superpixel errors, particularly given the very lightweight stem used in our experiments.
> >
> > | Method                          | FLOPs | Params | Accuracy |
> > |---------------------------------|-------|--------|----------|
> > | SPFormer-S/32                   | 1.2G  | 22M    | 76.4     |
> > | +STViT style                    | 1.1G  | 22M    | 68.5     |
> > | +STViT style with gradient flow | 1.1G  | 22M    | NaN      |
> >
> > Upon enabling gradient flow alongside [1]'s representation, we observed training instability which frequently resulted in NaN values. This instability corroborates our hypothesis that the gradient flow is intentionally disabled in [1], despite being inherently differentiable. Additionally, this outcome highlights the significance of implementing residual updates for superpixel representations to maintain training stability.
> >
> >
> > - [1] Varun Jampani, Deqing Sun, Ming-Yu Liu, Ming-Hsuan Yang, and Jan Kautz. "Superpixel Sampling Networks." ECCV 2018.
> > - [2] Huang, Huaibo and Zhou, Xiaoqiang and Cao, Jie and He, Ran and Tan, Tieniu. "Vision Transformer with Super Token Sampling." CVPR 2023.
> >
> > Through these distinctions, our method significantly shifts the paradigm of superpixel segmentation by reformulating it as sliding window cross attention within the SCA layer that enhances adaptability, accuracy, and the ability to learn from complex visual data efficiently.

---

### Review · Reviewer_RDXq · 2024-10-22

**Summary Of Contributions:**

This paper presents a new ViT class algorithm that leverages super pixel to build attention. It claims to be more flexible and adaptable using the segmentation level attention.

**Audience:**

Yes

**Broader Impact Concerns:**

The method is not elaborated in sufficient detail, and detailed comparisons with previous methods are missing.

**Claims And Evidence:**

No

**Requested Changes:**

The method needs to be clearly defined, and the novelty (difference) of this work with previous works needs to be further highlighted.

Minor: Better explain in plain text rather than place jargons, e.g. mathematically describe Moore Neighborhood rather than refer to the name without explanation.

**Strengths And Weaknesses:**

**Strengths:**

- The algorithm proposed in this paper is based on super pixel segmentations, hence comes with better explainability than patch based ViT. The experimental results have shown robustness to rotation and Occlusion as well.

- The overall structural of the paper is complete, with literature reviews on ViT and super pixel representations, and ablation studies showing the improvement of proposed methods.

**Weacknesses:**

- The super pixel learning part is not described in sufficient detail, making it hard for reader to comprehend how the pixel are grouped, and how much learning and engineering are involved to build the model. Specifically, $A_{ip}$, $S_{p}$ in EQ(1) are not defined nor discussed.

- Given the unclear core method definition, it is hard the core method, hence hard to validate the adaptation ability. How to ensure the total number of super pixel, or are they fixed at all? How to ensure there are no empty super pixel? How much offline and online computations are required - are they learned via clustering or gradient decent?

- The difference between super pixel learning in this paper with previous super pixel papers are not discussed, hence the novelty is hard to judge.

---

> ### Author Response · Authors · 2024-11-11
> **Response to Reviewer RDXq**
>
> **1. Clearer Notation and Method**
>
> We appreciate your feedback regarding the clarity of our method section. To address this, we have **thoroughly reorganized the method section** to ensure that the notations and procedures are clearly presented and easily understandable.
>
> **2. Number of Superpixels: Are They Fixed?**
>
> Yes, the number of superpixels is fixed in our framework. When given a pixel grid with dimensions $h \times w$, we employ a fixed stride $s=4$ to establish a superpixel grid of dimensions $s_h \times s_w$. This arrangement ensures that the total count of superpixels remains constant.
>
> **3. Ensuring Non-Empty Superpixels**
>
> We leverage the soft association provided by the softmax operations in equations (2) and (4) within the cross-attention mechanism to ensure that there are no empty superpixels. It is important to note that in our visualizations, which prioritize displaying only the most influential superpixels, some superpixels might appear to have no pixels directly assigned to them. However, these superpixels still acquire information from adjacent pixels via the SCA module.
>
> **4. Learning the Superpixels: Offline Computation?**
>
> As detailed in Section 3.2, our method reformulates the superpixel generation process through the introduction of the SCA module, which incorporates P2S and S2P sliding window cross-attention mechanisms. This module is fully differentiable and can receive gradient updates from both the updated pixel and superpixel features, promoting dynamic learning. Importantly, there is no offline computation required; superpixels dynamically emerge solely from interactions within the SCA module, even when using category annotations only.
>
> **5. Moore Neighbor Definition**
>
> Thank you for pointing out the need for a clearer explanation of the Moore neighbor and sliding window concepts. We have added Figure 2 in the revised manuscript to provide a visual aid and facilitate a better understanding of these definitions.
>
> **6. Comparing Superpixel Learning: This Paper vs. Previous Studies**
>
> Key Differences:
>
> 1. **Gradient Flow:**
>    The absence of gradient flow in [1][3] means their network remains unaware of the clustering process, inhibiting its ability to correct clustering errors, which is crucial for adaptive learning.
>
> 2. **Superpixel Segmentation Ambiguity:**
>    Given the inherent over-segmentation nature of superpixels, our method utilizes a multi-head cross-attention mechanism to produce a spectrum of possible superpixel segmentations. This approach is detailed in Section 3.2.3 of our paper and contrasts sharply with [1][2][3], which relies on a singular segmentation approach. Such a singular method may not capture the diverse contextual information present in complex images.
>
> 3. **Residual Update of Superpixel Representation:**
>    Unlike traditional methods and [1][2][3], our technique employs residual updates to the superpixel features, enhancing the training stability. This approach ensures gradual updates that build intelligibly upon prior iterations, facilitating better convergence and addressing common challenges like exploding or vanishing gradients.
>
> 4. **Heavy Backbone and Dense Annotation Requirements:**
>    [2] requires a full convolutional neural network (CNN) as input and dense annotations for learning superpixels, suggesting a dependency on substantial pre-existing data and substantial computational resources. In contrast, our approach with interleaved SCA layers enables the generation of superpixels from progressively refined context features augmented by ViT blocks without the need for dense annotations.
>
> - [1] Radhakrishna Achanta, Appu Shaji, Kevin Smith, Aurelien Lucchi, Pascal Fua, and Sabine Süsstrunk. "SLIC: Superpixels Compared to State-of-the-Art Superpixel Methods." TPAMI 2012.
> - [2] Varun Jampani, Deqing Sun, Ming-Yu Liu, Ming-Hsuan Yang, and Jan Kautz. "Superpixel Sampling Networks." ECCV 2018.
> - [3] Huang, Huaibo and Zhou, Xiaoqiang and Cao, Jie and He, Ran and Tan, Tieniu. "Vision Transformer with Super Token Sampling." CVPR 2023.
>
> Through these distinctions, our method significantly shifts the paradigm of superpixel segmentation by reformulating it as sliding window cross attention within SCA layer that enhance adaptability, accuracy, and the ability to learn from complex visual data efficiently.

---

### Review · Reviewer_FofP · 2024-10-26

**Summary Of Contributions:**

The paper introduces a novel vision transformer architecture that leverages a superpixel-based representation instead of the traditional patch-based approach. This superpixel representation is generated using the proposed SCA module, which enhances the interpretability of the vision transformer and improves its robustness to rotations and occlusions.

**Audience:**

No

**Claims And Evidence:**

No

**Requested Changes:**

See Weakness.

**Strengths And Weaknesses:**

Strengths:

1. It is interesting to integrate the superpixel representation in vison transformer.

2. The proposed method can achieve better performance than that of DeiT on classification and semantic segmentation.

Weakness:


1. The experiments are significantly insufficient, with only one baseline method included for the classification and segmentation tasks. The proposed approach is not compared against some of the latest methods. Additionally, a comparison with the similar work [1] is also essential to provide a more comprehensive evaluation.

2. The computational cost should be further justified by providing additional evidence such as actual running time, throughput or an analysis of time complexity.

3. In Table 5, the performance of SPFormer-B is lower than that of SPFormer-S. Can authors explain why SPFormer would achieve lower performance despite having more parameters in this task?


[1] Huang, Huaibo, et al. "Vision transformer with super token sampling." Proceedings of the IEEE/CVF conference on computer vision and pattern recognition. 2023.

---

> ### Author Response · Authors · 2024-11-11
> **Response to Reviewer FofP**
>
> **1. Baseline Choice and Focus on Representation**
>
> Thank you for your inquiry regarding the choice of baseline in our study. Our paper specifically emphasizes the basic representation used by ViT because our primary goal is to investigate the impact of modifications to the underlying data representation, rather than altering training strategies or attention patterns.
>
> We chose to maintain the core architectural framework of the ViT to isolate the effects of the representation changes. This methodology allows us to directly assess how these alterations enhance the model's efficiency, explainability, and robustness. By minimizing deviations from the standard ViT architecture, we ensure that any observed improvements can be attributed clearly and directly to the representation changes, facilitating a straightforward comparison.
>
> **2. Comparison with STViT [1]**
>
> Key Differences:
>
> 1. **Gradient Flow:** In [1], the superpixel sampling process does not allow for gradient flow. The absence of gradient flow means their network remains unaware of the clustering process, inhibiting its ability to correct clustering errors, which is crucial for adaptive learning.
>
> 2. **Superpixel Segmentation Ambiguity:** Given the inherent over-segmentation nature of superpixels, our method uses a multi-head cross-attention mechanism to produce a spectrum of possible superpixel segmentations. This approach is detailed in Section 3.2.3 of our paper and contrasts sharply with [1], which produces a single superpixel segmentation. This single approach may not fully capture the diverse contextual information present in complex images.
>
> 3. **Residual Update of Superpixel Representation:**
>    Unlike traditional superpixel methods and [1], our technique applies residual updates to the superpixel features. This approach helps stabilize the training process by ensuring that updates are gradual and build intelligibly upon previous iterations. Residual learning facilitates better convergence and mitigates common training issues like exploding or vanishing gradients, often seen in more complex or deeper networks.
>
> **Performance Comparison:**
>
> To empirically validate the theoretical differences highlighted, we incorporated [1]’s superpixel representation into our SPFormer-S/32 model for a direct performance comparison using the official released code. The substitution of our superpixel representation with that from [1] led to a notable decrease in model accuracy, as detailed in the table below. This underperformance is likely attributable to the disabled gradient flow in [1]'s method, which prevents the network from effectively correcting superpixel errors, particularly given the very lightweight stem used in our experiments.
>
> | Method                          | FLOPs | Params | Accuracy |
> |---------------------------------|-------|--------|----------|
> | SPFormer-S/32                   | 1.2G  | 22M    | 76.4     |
> | +STViT style                    | 1.1G  | 22M    | 68.5     |
> | +STViT style with gradient flow | 1.1G  | 22M    | NaN      |
>
> Upon enabling gradient flow alongside [1]'s representation, we observed training instability which frequently resulted in NaN values. This instability corroborates our hypothesis that the gradient flow is intentionally disabled in [1], despite being inherently differentiable. Additionally, this outcome highlights the significance of implementing residual updates for superpixel representations to maintain training stability.

---

> > ### Author Response · Authors · 2024-11-11
> > **Response to Reviewer FofP Continued**
> >
> > **3. Justification of Computational Cost through Time Complexity Analysis**
> >
> > To address the concerns regarding the computational cost of our approach, we present a detailed analysis of the time complexity associated with our method. This analysis aims to clarify the efficiency of our technique in computational terms.
> >
> > In the S2P cross-attention mechanism, each pixel $i$ evaluates its association with its neighboring superpixels, denoted as $\mathcal{N}_i$, which includes $n = r^2 = 9$ superpixels. Therefore, the time complexity for S2P cross-attention is $O(nhw)$, where $h$ and $w$ are the height and width of the pixel features, respectively.
> >
> > Similarly, the time complexity for P2S cross-attention can be elaborated as follows: considering a stride $s=4$, the complexity is $O((rs)^2 s_h s_w)$. By substituting the radius value, we get the overall time complexity for P2S as $O(n s^2 s_h s_w)$. It's important to note that $s_h s_w$ represents the dimensions of the superpixel grid, which vary based on the stride and image dimensions.
> >
> > This shows that our method's computational complexity is linear with respect to the number of superpixels, contrasting with the quadratic complexity observed in the later ViT layers.
> >
> >
> > **4. Performance Discrepancy Between SPFormer-B and SPFormer-S in Table 5**
> >
> > Table 5 evaluates superpixel quality in a zero-shot setting, where the models are trained solely on ImageNet with category annotations. The superpixels generated by SPFormer-B may align more closely with the ImageNet distribution, which does not necessarily transfer effectively to the Pascal VOC dataset.
> >
> > Additionally, SPFormer-B uses 3 heads, while SPFormer-S uses 2 heads. Our method enables multiple superpixel segmentations, and in this analysis, we simply average across multiple heads to compute the metrics. This approach is suboptimal, as the same superpixel $p$ may correspond to different semantic parts across heads, leading to potential performance issues, especially with more heads. As mentioned in Sec. 4.3.2, effective feature extraction in our model is achieved if even a single head accurately identifies a superpixel. This experiment illustrates the model's ability to learn superpixels without dense annotations, despite the limitations of this setup.

---

### Review · Reviewer_dhve · 2024-10-27

**Summary Of Contributions:**

The authors propose a Superpixel Cross Attention operation to leverage a superpixel representation within Vision Transformers. They show that this leads to relatively large gains in accuracy (1%) on ImageNet-1k classification for a relatively small increase in FLOPs. The method also results in substantially better (4%) zero-shot performance on datasets such as COCO and Pascal VOC 2012. Interestingly, they show enhanced robustness to rotations up to +7% using the superpixel representation.

**Audience:**

Yes

**Claims And Evidence:**

Yes

**Requested Changes:**

Section 3.2, and to a lesser extent 3.1, could include a clearer explanation of the technique. This might be as simple as defining symbols such as N, W, n, r. The paper assumes familiarity with previous work on superpixel representations, and could reach a wider audience by being more self-contained. Further, the architecture is somewhat unclear, with details interspersed in Sections 3.3, 4.1, 4.2. The architecture details/SCA details should be gathered into clear sections rather than mixed with experimental details, IMO (this can work sometimes, but the overall view of the architecture is somewhat confusing in this case). Please also see Weaknesses, though I don't really expect every point to be addressed.

I anticipated that this paper would be solely about changing the input representation, but I think the SCA layers are actually dispersed throughout the ViT architecture. If the authors want to present a more input-representation-centric perspective, they might be interested in mentioning works such as [0, 1, 2].

Overall, I think this is a neat contribution, but the reach will be limited by lack of clarity / reproducibility.

[0] Adaptive Input Representations for Neural Language Modeling, Baevski and Auli (more for language than vision, admittedly)
[1] Patches are all you need? Trockman and Kolter (there is much discussion of the patch representation)
[2] MetaFormer Is Actually What You Need for Vision, Yu et al (might show some support for the importance of input representation vs. architecture)

**Strengths And Weaknesses:**

Strengths
- Fairly large improvements to ImageNet top-1 accuracy with minimal extra compute (especially nice for DeiT-S comparisons).
- Quite large improvements for robustness to rotation (thought this was most interesting).
- Enhanced explainability via superpixel representation.
- Notable improvements to object-level and part-level classification on Pascal VOC 2012 (zero shot, trained only on ImageNet) compared to solely-patch-based ViT.

Weaknesses
- The method isn't entirely clear from Sec 3.1. It would be nice to define r and n more generally, surely these aren't fixed at r = 3 and n = 9 (n is the number of superpixels, I assume).
- Not clear how to construct N and W (neighboring superpixels, superpixel local window) in Sec. 3.1 -- maybe this is more obvious if you're familiar with previous work on superpixels, but since this is the key contribution of the paper it would be good to be more precise.
- Not sure what subscript ip is in Eq. 1 -- I'm guessing p is a (h,w) pair, and N_i is a set of such pairs?
- Somewhat confusing that P2S is presented as a single equation (2) and S2P as two equations (3-4).
- On that note, since the SCA module is the key contribution, would be nice to have its exact functional form specified (I guess I'm inferring from Fig. 2 that there are two activations S and I which are maybe concatenated later?)
- When you say iteratively, you mean there are multiple S2P/P2S layers? Otherwise, unclear what is meant by "iterations"
- In Sec S3.3, the fact that superpixel features come from a 1x1 conv followed by 4x4 average pooling seems quite important. I guess broadly speaking, the actual form of the architecture is a bit mysterious based on the paper, and could be more clearly laid out.
- For example: it seems by the end of 3.3 that SCA isn't just part of the stem, but actually interleaved into the ViT. This could be more clear/precise.
- Sec 4.1 contains more (crucial?) implementation details despite being in the Experiments section.
- In 4.1, kind of sounds like we you still use 4x4 patches before processing superpixels (which is okay, just should be lifted into 3.3 / an architecture section)
- (Fairly generic) experiment details are mixed into novel architecture details in Sec 4.1, these should be organized better
- More architecture changes within results in 4.2.1, adding several 3x3 stride-2 convs unexpectedly -- and this seems quite crucial to performance...
- There is generally quite a bit of discussion about the efficiency of this representation, but in the tables SPFormer seems to use more FLOPs for relatively small accuracy gains -- I know +1% is quite a bit for ImageNet, but it's also something like 10% more FLOPs. I think this is fine, but maybe deserves explicit discussion (e.g., how much accuracy would we expect to gain from increasing FLOPs by this much through another technique). This is minor, I appreciate the technique regardless.

---

> ### Author Response · Authors · 2024-11-11
> **Response to Reviewer dhve**
>
> **1. More General Definition of $r$ and $n$**
>
>  We appreciate your suggestion regarding the neighborhood radius $r$. While we initially used ( $r = 3$ ) based on common practices in the field of superpixel segmentation, we acknowledge that a broader range of values could be more illustrative and applicable in various scenarios. Consequently, we have revised Section 3.1 of our manuscript to include a more general definition.
>
>
> **2. Clearer Definition of Neighboring Superpixels and Superpixel Local Window**
>
> Thank you for pointing out the need for clarity in defining neighboring superpixels and the concept of a superpixel local window. To address this, we have introduced a new figure 2 which visually depicts these concepts, ensuring the manuscript is self-contained and more comprehensible. This image should aid readers in visualizing the arrangement and interaction of superpixels relative to a focal pixel.
>
> **3. Definition of the Subscripts $i$ and $p$**
>
> To clarify, the subscript $i$ pertains to a specific pixel located within a grid of pixels that is characterized by its dimensions $(h, w)$. Correspondingly, we introduce the concept of a superpixel grid defined by the dimensions $(s_h, s_w)$, where the subscript $p$ denotes an individual superpixel within this grid.
>
> Additionally, $N_i$ represents the set of superpixels contained within the local window associated with pixel $i$. For a visual representation and further details, please refer to the added fig. 2.
>
> **4. Presentation of P2S and S2P Formulas**
>
> Thank you for the suggestion. We wish to clarify that Equation (4) is a specialized variant of Equation (1). To elucidate the physical implications, we have revised Section 3.2 accordingly.
>
> **5. Output of the SCA Layer**
>
> In determining the output of the SCA layer, we use the final iteration of pixel features and superpixel features, rather than concatenating all intermediate outputs.
>
> **6. Iterations in the SCA Layer**
>
> Within the SCA layer, the S2P/P2S attentions are applied through two iterations. In each iteration, we recalculate the association $A$ using the updated pixel and superpixel features from preceding iterations. This is inspired by traditional superpixel algorithms.
>
> **7. Initialization of Superpixel Features**
>
> To increase the feature dimension from pixel features $c$ to superpixel feature dimension $s_c$, a 1x1 convolution is used. A 4x4 average pooling retains the count of superpixels equal to the number of patches in the original ViT model, minimizing extra learnable parameters for a fair comparison. Section 3.3 has been revised to provide clearer details of the architecture.
>
> **8. Reorganization of the Architecture Section**
>
> Thank you for your suggestions. We have **completely reorganized the Methods section** to enhance clarity and ease of understanding.
>
>
> **9. Additional Parameters and Efficiency**
>
> Thank you for your insightful comments on the computational efficiency of our method. In the main paper, our goal is to keep changes minimal, focusing differences primarily on the internal representation. To address the concern and provide a clearer comparison, we adjusted our SPFormer-S model by removing one ViT block, reducing its FLOPs to 4.8G. This modification allowed the model to achieve an 80.9% top-1 accuracy on ImageNet, demonstrating an improved balance between computational cost and performance.
>
> **10. Input Representation**
>
> Thank you for the insightful references. Rather than merely altering the input representation, our goal is to enhance the foundational representation across the ViT. Through the interleaved SCA modules, we achieve ongoing improvements in the superpixel representation.

---

### Author Response · Authors · 2024-11-11
**Global Response**

We sincerely thank all reviewers for the constructive feedback, which is helpful to improve the quality of our paper. Firstly, we follow the reviewers' suggestion to revise the paper accordingly, where the modified parts are marked in red. Then, regarding detailed concerns, we address them in our response to each reviewer.

---

### Decision · Action_Editor_ciKn · 2024-12-13

**Recommendation:** Accept with minor revision

**Comment:**

The reviewers' scores are somewhat mixed. Although the authors have added experiments in their rebuttal by incorporating [1]'s superpixel representation into the SPFormer-S/32 model, they did not directly compare it with [1] or other related methods. This direct comparison is crucial for assessing the effectiveness of the proposed method, especially in the context of existing approaches that address similar challenges. Additionally, Table 1 appears insufficient in demonstrating the algorithm's efficiency advantages, particularly when compared to other similar methods.

Despite the concerns mentioned above, reviewers agree that this work introduces a novel architecture that integrates super-pixel representations to address the limitations of fixed partitions in traditional Vision Transformers (ViT). Therefore, this paper has value for publication, but it would be beneficial to address the remaining concerns to make the claims more convincing.

**Audience:**

Some individuals in TMLR's audience would likely be interested in the findings of this paper. The integration of super-pixel representations to address fixed partitions in traditional Vision Transformers is interesting.

**Claims And Evidence:**

The claims made in the submission are partially supported but not convincingly demonstrated. The primary concern arises from the lack of comparisons with existing methods that address similar challenges in Vision Transformers, such as shifting local windows or deformable self-attention. This gap makes it difficult to assess the proposed method's novelty and effectiveness accurately. Furthermore, the claim of improved computational efficiency is not adequately justified, as the evidence provided is insufficient to support this claim.